# Microfluidic Approaches for Affinity-Based Exosome Separation

**DOI:** 10.3390/ijms23169004

**Published:** 2022-08-12

**Authors:** Eike K. Theel, Sebastian P. Schwaminger

**Affiliations:** 1Bioseparation Engineering Group, School of Engineering and Design, Technical University of Munich, Boltzmannstraße 15, 85748 Garching bei München, Germany; 2Division of Medicinal Chemistry, Otto Loewi Research Center, Medical University of Graz, Neue Stiftingtalstraße 6, 8010 Graz, Austria

**Keywords:** exosomes, affinity separation, microfluidic chamber, purification, extracellular vesicles, µTAS, LOC

## Abstract

As a subspecies of extracellular vesicles (EVs), exosomes have provided promising results in diagnostic and theranostic applications in recent years. The nanometer-sized exosomes can be extracted by liquid biopsy from almost all body fluids, making them especially suitable for mainly non-invasive point-of-care (POC) applications. To achieve this, exosomes must first be separated from the respective biofluid. Impurities with similar properties, heterogeneity of exosome characteristics, and time-related biofouling complicate the separation. This practical review presents the state-of-the-art methods available for the separation of exosomes. Furthermore, it is shown how new separation methods can be developed. A particular focus lies on the fabrication and design of microfluidic devices using highly selective affinity separation. Due to their compactness, quick analysis time and portable form factor, these microfluidic devices are particularly suitable to deliver fast and reliable results for POC applications. For these devices, new manufacturing methods (e.g., laminating, replica molding and 3D printing) that use low-cost materials and do not require clean rooms are presented. Additionally, special flow routes and patterns that increase contact surfaces, as well as residence time, and thus improve affinity purification are displayed. Finally, various analyses are shown that can be used to evaluate the separation results of a newly developed device. Overall, this review paper provides a toolbox for developing new microfluidic affinity devices for exosome separation.

## 1. Introduction

Extracellular vesicles (EVs) are, nowadays, one of the most promising biological constructs [1]. Previously considered as cellular waste [2,3], new research suggests that their specific composition can be used to determine the status of cancer, auto-immune or cardiovascular diseases from mainly non-invasive liquid biopsies [4,5,6,7,8]. Moreover, it is possible to turn them into targeted drug delivery systems for theranostic application with biochemical engineering methods [9,10,11,12,13,14,15,16,17,18]. To exploit these diagnostic and therapeutic opportunities, efficient methods for separating EVs from biological solutions must be found. The limitations and drawbacks of conventional separation methods have been overcome in recent years using new microfluidic systems and highly efficient separation principles, such as affinity binding. Therefore, this practical review focuses on the methodologies that are currently available to create new devices for EV separation using state-of-the-art technologies.

To separate EVs, it is important to know how they are defined and which properties they possess. The International Society of Extracellular Vesicles (ISEV), which accumulates all research results about EVs, defines them as “particles naturally released from the cell that are delimited by lipid bilayer and cannot replicate” [19,20]. These lipid bilayer vesicles can further be divided by their biogenesis and properties into exomeres (Ø < 50 nm), exosomes (Ø 30–150 nm), ectosomes or shedding microvesicles (Ø 100–1000 nm), apoptotic bodies (Ø 1000–5000 nm), migrasomes (Ø 500–3000 nm) and large oncosomes (Ø 1000–10,000 nm) [21]. Although more knowledge has recently been accumulated on all subtypes of EVs, current research is mainly focused on exosomes as a promising EV species [1,22]. The ISEV also recommends the use of the term small extracellular vesicles (sEVs) to describe exosomes.

Exosomes are formed inside the cell by inward budding of the endosomal membrane, and therefore represent a snapshot of the current status of the donor cell. After an intermediate state as intraluminal vesicles within multivesicular bodies, they are exocytosed. These vesicles are then referred to as exosomes. Outside the cell, exosomes can act as intercellular transporters, carrying their protected cargo to other cells [23]. Depending on the location of the cell, it is, thus, possible to isolate exosomes from nearly all accessible body fluids, such as systemic body fluids (e.g., blood, breast milk, follicular fluid, seminal fluid, serum, urine) and proximal body fluids (e.g., cerebrospinal fluid, saliva, sweat, tears) to analyze their composition [24].

The ability to determine the current status of a cell using exosomes via minimal to non-invasive liquid biopsies has great potential, especially for cancer diagnostics. However, the exploitation of this potential is hindered by several factors that complicate the separation of exosomes. One factor is the heterogeneity of the sites of origin, which significantly alters the membrane composition and cargo of exosomes. A second factor includes the impurities that interfere with efficient purification depending on the biofluid. Two examples are nucleic acids and lipoprotein particles (LPs) [25]. LPs are present in human serum and have a similar size and density range to exosomes [26]. Therefore, the separation of exosomes and LPs with methods that utilize these properties is difficult. Thirdly, the time-related degradation of exosomes, also referred to as biofouling, is challenging [27]. This makes a fast purification process necessary to achieve a high yield. Lastly, the general physical properties of the exosomes should be mentioned. They have a buoyant density of 1.10–1.14 g cm^−3^, an overall negative charge and, as already mentioned, a diameter in the nanometer range. Thus, there are high demands on the sample processing and measuring instruments.

Besides the properties of the exosomes that make separation difficult, their unique molecular composition (biomarker) can be utilized to isolate and analyze them [12,14,28,29] (Figure 1). Generally, exosome biomarkers are located on the surface or inside as so-called cargo and can either be general markers or disease associated [30]. Several of these biomarkers are known and published by the ISEV in their annual report or in databases, such as the ExoCarta [31]. The surface markers can be used to capture intact exosomes via affinity binding mechanisms that recognize specific sequences [32]. Proteins and lipids that can be found in the cellular membrane can also act as markers for recognition [33,34,35]. These biopolymer markers can be enriched by up to multiple magnitudes in the exosome membranes in comparison to their occurrence in donor cells [36]. Typical examples of these markers are tetraspanins and cholesterol [36,37,38,39]. Thus, the separation strategy is strongly dependent on the donor cells that the exosomes originate from [40]. For this purpose, the following Table 1 and Table 2 provide guidance to define the initial targets for affinity separation processes. In particular, the cell differentiation factors CD9 and CD63 are frequently used for specific exosome binding [34,38,41,42,43]. However, in a 2020 paper by Hoshino et al., it was found that CD63 may not occur on all exosomes [28]. Instead, two other proteins, moesin (MSN) and b2-microglobulin (B2M), were proposed as general biomarkers for exosomes [28,44,45]. As exosomes can be used as vehicles, they can transport molecules as cargo that are particularly suitable for cancer diagnostics [46,47]. In particular, the contained mRNA, miRNA, and other (long) non-coding RNA (lncRNA) strands have been used as biomarkers in recent years [48]. These can be evaluated after lysis of the exosomes and subsequent quantitative polymerase chain reaction (qPCR) to identify deviation from healthy cells [49].

## 2. Separation Mechanisms

First, general considerations for designing an exosome separation device are discussed. Second, factors that need to be considered when choosing a separation method for a specific application are highlighted. Usually, several purification principles need to be combined for the successful purification of exosomes due to the afore mentioned challenges. Hence, different principles that are already successfully used for the separation are presented in the last part of this section. Among the reviewed separation principles, certain approaches that exploit the affinity between biomarkers and their binding partners have been able to achieve particularly high purities of exosomes [51]. Thus, the potential of affinity separation will be discussed in more detail. In recent years, affinity separation principles were combined with a modern format for separation and detection to produce microfluidic devices. Microfluidic devices have multiple advantages, such as their ability to reduce sample and buffer volumes significantly. This is only one reason why their use in the separation of exosomes has increased exponentially since 2014 [50]. Therefore, this review specifically focuses on how such microfluidic devices can be manufactured and designed for optimal exosome separation and purification. The combination of the following chapters creates a toolbox that can be used as a basis for the construction of novel devices for the separation of exosomes.

### 2.1. General Considerations for the Selection of an Exosome Separation Process 

Due to the challenging factors in exosome separation, complete isolation of exosomes from their respective biofluids is considered by the ISEV as an “unrealistic goal” [19]. Therefore, their guidelines will be followed in the subsequent sections and the term “isolation” [52], which is frequently used in the literature, will be avoided. Instead, the terms separation, concentration or enrichment will be used synonymously throughout this review. The first step in developing a new separation method is often based on a review of previously developed techniques. However, despite multiple attempts, a comparison of the many different separation procedures for exosomes remains challenging [51]. The ISEV proposes a categorization of the different methods in terms of recovery and specificity into four classes, whereby the highest class (4: high recovery and high specificity) has not yet been reached by any separation method. A great comparison for exosome separation has been provided by Gulei et al. who provided a compact comparison for different strategies, assigning one to three “+” signs for the parameters yield, purity, time, and costs [51]. They highlight density gradient centrifugation as the best process for all four parameters, while other techniques, such as ultracentrifugation, ultrafiltration, precipitation, magnetic cell sorting, immune-affinity techniques, affinity assays, acoustic nanofilters, nanopillars and ExoChips, show disadvantages in yield and costs [51]. In contrast to this fast evaluation of techniques, the literature reviews on EV isolation by Cheng et al. [50] and Ding et al. [45] contain detailed tables with the description of the advantages and disadvantages or limits of the individual separation processes. Wang et al. also give a comprehensive overview of the different separation systems for exosomes, evaluating filtration, immuno-magnetic separation, centrifugation and precipitation [53]. Patel et al. compared four different commercially available kits with the separation of exosomes by ultracentrifugation in detail [54]. In a recent review paper by Hassanpour Tamrin et al., the specific parameters for the separation process are highlighted. Sample volume, required time, size range, recovery rate and purity for multiple microfluidic-based exosome separation methods are described in detail for the isolation of exosomes [52]. All the review papers mentioned have been published between 2018 and 2021, which emphasizes the great interest in the ongoing research and the need for purification technologies. This also shows the dynamics in the field and an urge for an overview of the existing separation methods. As more and more separation methods are published, it is feasible to carry out a multidimensional analysis of the parameters to find the method that fits the goals of the experimenter the best [55]. Parameters can be categorized into economic and engineering parameters. Economic parameters are time, costs (samples and equipment), technology readiness level (TRL) and qualification of the employees. Engineering parameters, such as purity/selectivity, yield, recovery, volumetric throughput, sample consumption, robustness and integrity of the exosomes, must be evaluated for each separation method. Unfortunately, these parameters are not always easily accessible. This highlights the need for a uniform database where the above-mentioned parameters for every new method are provided by the developers.

### 2.2. Separation Systems Depending on the Applications of Exosomes

Another important point when comparing separation processes for exosomes is the target application. The separation methods often described as “conventional” are derived from fundamental academic research about EVs, while only a few industrial methods exist [56]. The primary goal of this application is to obtain exosomes with high purity using special laboratory equipment (e.g., ultracentrifuges) to gain a deeper understanding of exosomes. Due to the high potential of exosomes for cancer diagnostics, methods have simultaneously been developed that are suitable for the application of separation methods for point-of-care (POC) diagnostics [57,58]. The overall goal of POC diagnostics is to have fast and simple testing systems that can be used everywhere with portable instruments [59]. For exosome analysis, it would be ideal if fresh liquid biopsy samples from patients could be analyzed quickly and on-site [29]. Sensors and detectors are often developed for this application. Requirements for POC devices are that no specialized or difficult to operate laboratory equipment is used and that the analysis can be carried out in a short time in an “unclean” environment. Such requirements can be met by the development of closed lab-on-a-chip (LOC) or micro-total-analysis systems (µTAS). Their multidisciplinary development goes hand in hand with the improvement of miniaturized detectors and novel manufacturing processes for microfluidic devices. Some microfluidic devices have shown higher purities in a shorter time compared to the conventional separation methods [60]. This and the other advantages already mentioned of microfluidic devices makes them interesting even for fundamental research on exosomes. A third possible application for exosome separation is the recovery of exosomes for ex vivo modification (analogously to the therapy with CAR-T cells) or the enrichment of artificially produced mimetic nanovesicles to generate targeted drug delivery systems [61,62]. Due to their small size, great biocompatibility, protection of endogenous cargos, and almost no immunogenicity, they have great potential for the theranostic therapy of cancer [12]. The aim of using exosomes for theranostic applications is to achieve the highest possible purity to facilitate the modification and to make them safe for (re-)injection into patients. Due to the variety of different applications, a disadvantage of a separation method for one application can become an advantage for another method or in another format. To decrease the chance of a biased selection, the following conventional separation methods are presented in a neutral manner. 

### 2.3. Conventional Separation Methods for Exosomes

Conventional methods are methods that are carried out with commonly available laboratory equipment and have already been reproduced several times. Although the focus of this review is on novel separation methods using separation formats, such as microfluidic devices, the concepts of conventional methods also have the potential to be adapted within these newer formats and need to be considered. Moreover, these methods are often used as references when evaluating new methods [63]. All separation processes are based on one of the following three separation principles or a combination of them: particle shape (size, steric hindrance), field behavior (gravitation, electric and magnetic fields), and chemical properties (affinity binding, precipitation).

#### 2.3.1. Centrifugation-Based Methods

The most used method for the separation of exosomes is the separation according to their density and size by means of differential centrifugation (DC) [64]. In the first centrifugation step, the cells and apoptotic fragments can be removed at relative centrifugal factor (RCF) values between 300× *g* and 2000× *g*. In the second step, the RCF is increased to 10,000× *g* to separate larger EVs from cellular metabolites and protein aggregates. An additional ultracentrifugation (UC) step at 100,000× *g*–200,000× *g* must be applied to pellet exosomes, which can then be washed with PBS buffer [21]. During the process, the samples are usually cooled to 4 °C to reduce biofouling. An extension of DC is the density gradient ultracentrifugation (DGC). Here, sucrose or iodixanol is used as a high-density medium in a gradient solution or as a cushion to improve the separation [65,66].

#### 2.3.2. Ultrafiltration Methods

In ultrafiltration, particles between Ø 2 and 100 nm can be separated; thus, this is also the case for exosomes. There are generally two modes of operation, which are s follows: the filtration can operate as a dead-end process or as a tangential flow (TF) process. Dead-end filtration can also be combined with centrifugation, in which a filter is installed in a centrifuge tube. TF methods reduce the clogging of the filter medium and allow continuous separation [67]. In a paper published in 2021 by Chen et al., the TF method with two membranes was combined with a harmonic oscillator. This further improved the purification [25].

#### 2.3.3. Precipitation Methods

Another common method of exosome separation is precipitation with polyethylene glycol (PEG). It may be feasible to first centrifuge the sample to remove cell residues. Then, PEG and NaCl are mixed with the supernatant so that a total of 10% *v*/*w* PEG and 75 mM NaCl is reached. After an incubation time of at least 8 h, the exosomes can be concentrated by centrifugation. The PEG and other impurities can be removed by a washing step and further ultracentrifugation. Since PEG is a polymer that occurs in different lengths, studies were conducted to determine the optimal length. In the 2018 paper by Ludwig et al., PEG 6000 is suggested as particularly suitable [68]. Multiple ready-to-use kits that use this separation principal are commercially available and are as follows: miRCURY Exosome Kit from QIAGEN (Venlo, The Netherlands), ExoQuick-TC from System Biosciences (Palo Alto, CA, USA), Total Exosome Isolation Kit from Thermo Fischer Scientific (Waltham, MA, USA) and Exo-Prep from HansaBioMed Life Sciences (Tallin, Estonia).

#### 2.3.4. Field Flow Fractionation Methods

Field flow fractionation (FFF) utilizes the effect of vertically acting field forces on particles in a continuously flowing carrier solution to sort them. These permanent force fields can be magnetic, thermal, electric, or gravitational. An extension of the FFF is the asymmetric flow field flow fractionation (AF4). While the sample flows through a channel, an additional crossflow through a semipermeable wall is applied. This crossflow forces all the particles away from the semipermeable membrane. Due to their higher diffusion rate, and the resulting flow profile in the channel, the smaller particles are eluted earlier. Zhang et al. used this technique to separate exosomes with the commercially available Eclipse module from Wyatt Technologies (Santa Barbara, CA, USA) [69].

#### 2.3.5. Chromatographic Methods

Chromatography describes all the processes that apply a so-called stationary phase, which can be used to separate a mixture of components that are carried by a mobile phase [70]. Mainly size exclusion chromatography (SEC) is used for the separation of exosomes. In SEC, the smaller molecules can enter the stationary phase and are, thus, retained. The molecules that are larger than the pores of the stationary phase remain in the mobile phase and are eluted earlier. Some companies have already produced specialized SEC columns for the purification of exosomes, such as the qEV10 from IZON (Christchurch, New Zealand) or the PURE-EVs from HansaBioMed Life Sciences (Tallin, Estonia). In addition, established SEC media, such as the Superdex 200 Increase from Cytiva (Uppsala, Sweden), can be utilized for the exosome separation. In recent years, multiple extensions to the basic SEC principles have been developed. For example, in 2021, Li et al. proposed a negative isolation strategy for EVs (NIEV) [71]. Thereby, protein impurities are irreversibly bound to graphene oxide inside perforated polyether sulfone particles, while the exosomes cannot enter the pores and remain unbound on the outside. The SEC can additionally be extended with binding properties. One example is the Capto Core 700 resin from Cytiva (Uppsala, Sweden), which also contains multimodal ligands and leads to improved exosome purification [72]. Another chromatographic method that is used for the purification of exosomes is affinity chromatography (AC). In this method, a functional group is attached to the stationary phase, which can selectively bind to a structure on the target molecule. This is utilized to separate the molecule from its mixture. After a buffer change, the bound molecule can be released. The goal of affinity binding can either be to remove a specific impurity or to bind to the target molecule. It is possible to separate exosomes with different combinations of stationary phases and functional groups. For example, Wang and Turko used a heparin-sepharose matrix to selectively bind exosomes [73]. Another AC approach was performed by Bellotti et al. with an immobilized metal affinity (IMAC) CIMultus IDA column from Sartorius (Göttingen, Germany) and prior multimodal-extended SEC. Folate receptor α (FRα) was selected as the biomarker for exosomes. In order to use the IMAC separation, the natural FRα was modified with a polyhistidine-tag (His-tag) via genetic modification of the exosome-producing cell line (HEK293). Thus, the His-tagged exosomes bind to the copper ions of the IMAC and can later be eluted with an imidazole-containing buffer [72].

#### 2.3.6. Affinity Binding-Based Separations

Affinity methods can generally be categorized by the matrix onto which the affinity ligand is coupled. Besides the immobilization of affinity ligands on spherical polymer beads or monoliths, as in AC, they can also be attached to plates, magnetic particles, and membranes, or be soluble (see Figure 2). In the free ligand approach, the target molecules are first labelled by binding to ligand molecules in solution. In the following, either the ligand itself can be bound; thus, the target structure can be separated, or a signal reaction with the ligand can be used to detect the target structure. In the immobilized ligand approach, there is no labelling of the exosomes with free ligands, but direct separation or detection by binding to an immobilized ligand. Possible ligands are proteins, antibodies, and aptamers. The latter are synthetic single-stranded DNA or RNA sequences. They have similar binding properties to antibodies, but are more stable, less expensive and show less variance in their production [74].

To find the optimal combination of ligands and target structures for affinity separation of exosomes, it is important to compare their binding strength. The strength has high impact on the selection of the flow throughput of an affinity separation device. A common value to do so is the equilibrium dissociation constant *K**_eq,d_*. It describes the ratio of the dissociation rate (*k**_d_* = *k**_off_*) to the absorption rate (*k*_*a*_ = *k*_*on*_) of a ligand and its binding partner and represents the reciprocal value of the equilibrium affinity constant *K**_eq,a_* [75]. Therefore, it is possible to quantify the strength of the affinity bond with the *K**_eq,d_*-value—the lower the value, the higher the affinity. Typical values for antibody-antigen systems are in the range of *K**_eq,d_* = 10^−8^–10^−10^
*M* and for the extremely strong streptavidin-biotin system, *K**_eq,d_* = 10^−15^
*M* [74]. Since the values differ greatly depending on the measuring instrument used, it is not feasible to only rely on literature values for the comparison of different ligand-target systems. Furthermore, there are strong deviations in the binding performance of ligands from different sources and even lot-to-lot deviations (especially with antibodies). Thus, it is recommended to measure the *K**_eq,d_*-value of the comparing ligands-target systems manually while setting up the experiment [76]. Multiple methods have been developed for this purpose, such as radioligand binding assay (RBA), surface plasmon resonance (SPR), fluorescence energy resonance transfer method (FRET), affinity chromatography, and isothermal titration calorimetry (ITC) [77]. The SPR method is utilized by the commercially available Biacore instrument from Cytiva (Uppsala, Sweden). Even novel microfluidic devices are being developed for the fast determination of the *K**_eq,d_*-value. One example is the Auto-affitech device by Guo et al., which used microbeads for the determination of *K**_eq,d_*-values of aptamer-protein and antibody-antigen systems [78]. A challenge in the separation of exosomes using affinity methods is to dissolve the binding between the ligand and target structure to receive an enriched exosome solution. One way to dissolve the bond is to chemically alter the environmental condition by changing the salt concentration or pH-value. This is a well-established method in protein A chromatography. Furthermore, the bond can be irradicated by competitive elution. An example for this is the competitive displacement of biotin-labelled proteins from streptavidin columns using free biotin. Another biochemical method is the enzymatical cleavage of a part from the immobilized ligand. In addition, detachment can be provoked by physically changing the environmental conditions. This includes temperature or electrical impulses. Affinity kits based on different working principles are commercially available. For example, the ExoELISA Complete Kit from System Biosciences (Palo Alto, CA, USA) is an enzyme-linked immunosorbent assay (ELISA), with CD63 as the target exosome marker. The Exosome Human Isolation Kit from Thermo Fischer Scientific (Waltham, MA, USA) uses magnetic particles (Dynabeads) that specifically bind to CD9. In the ExoEasy Maxi Kit from Qiagen (Venlo, The Netherlands), the affinity ligands are bound on a complex membrane in a centrifugation tube. In addition to the products that are already commercially available, scientific publications are continuously appearing that extend or combine the existing working principle. For example, Boriachek et al. developed gold-loaded ferric oxide nanocubes functionalized with CD9 or CD63 antibodies [79]. The nanocubes enabled the separation of exosomes with a magnet without prior purification. Following the exosome separation, an integrated ELISA-based detection step was utilized for the direct detection of exosome subtypes [79]. Another example for the separation with magnetic nanoparticles involves the extracellular vesicle total recovery and purification (EVtrap) particles developed by Wu et al. They utilize “beads modified with a combination of hydrophilic and lipophilic groups that have a unique affinity toward lipid-coated EVs” [80]. In addition, approaches based on new working principles and matrices are being developed. Akbarinejad et al. developed an electrospun cloth to which gold particles and an aptamer against CD63 were bound [81]. After binding to the aptamer, it is possible to release the exosomes non-destructively by an electrochemical impulse.

## 3. Microfluidics—How to Build an Affinity Exosome Separation Chip

Microfluidic devices for the separation of exosomes have several advantages over conventional separation methods. Due to miniaturization, the required sample and buffer volumes are reduced. Reactions with immobilized ligands on the channel walls of a microfluidic device benefit additionally from the large surface-to-volume ratio. An increase in the reaction rate leads to a reduction in the separation or analysis time [82]. A reduction in the process time reduces the time-dependent degradation of exosomes, and thus reduces the variation in the separation results. By copying the process path several times in one device so that it can be run in parallel, this effect can even be multiplied [60]. Another advantage of the small distances in microfluidic devices is that thermal transport is faster than in larger scale operations. The improvements in the miniaturization made it possible to integrate multiple operations into one microfluidic device [83]. This enables an easier automation of the sample processing and reduces the amount of equipment, labor, and risk of cross-contaminations. These advantages can lead to high reproducibility and accuracy [84]. Overall, microfluidic devices are ideally suited for applications in a POC environment [85]. Microfluidic devices are defined as devices for the manipulation of fluids at the microscale level that utilize channels with diameters between 10 and 200 µm. While the field of microfluidic devices once started with capillary systems for gas chromatography and electrophoresis, the focus of this technology nowadays lies more on the chip format [86]. Examples of this format are microreactors, organ-on-a-chip, and LOC or µTAS devices [57]. Multiple manufacturing methods have been developed and optimized for different materials. Typical processes are based on mold manufacture, including mechanical (micro-cutting; ultrasonic machining), energy-assisted methods (electro-discharge machining, micro-electrochemical machining, laser ablation, electron beam machining, focused ion beam (FIB) machining) and traditional micro-electromechanical systems (MEMS) processes [87]. In this review, we focus on the manufacturing methods labelling, molding, and especially 3D-printing. These approaches do not require excessive amounts of equipment or clean-room conditions and are, therefore, better suited for low-cost POC devices. It is crucial for successful manufacturing to find the optimal material for the target application. Important properties for the selection and comparison of materials are as follows: durability, ease of fabrication, transparency, biocompatibility, chemical compatibility with the implied reagents, meeting the temperature and pressure conditions needed for the reaction, and the potential of the surface functionalization [88]. Commonly used materials are glass, metal, silicone, low temperature cofired ceramics (LTCC) and polymers. Polymers have become increasingly popular in recent years, thanks to their low cost and the ease with which they can be used to build microfluidic devices [89]. Common representatives of this group are polymethylmethacrylate (PMMA), copolymers and cyclo-olefin polymers (COPs/COC) and polydimethylsiloxane (PDMS). A comparison of the different materials can be found in the recent paper of Niculescu et al. [88].

### 3.1. Manufacturing Methods of Microfluidic Devices

LTTC tapes, PMMA and COC can be used in a manufacturing process called laminating (see Figure 3). This technique is characterized by multiple, differently cut layers that are piled up and bonded together [90]. The combination of these patterns in different layers creates microchannels. To cut the layers, a knife plotter or a laser cutter can be used [87]. Laser cutters are expensive but can realize high accuracies of ± 25 µm [91]. The cutting pattern for the different layers can be designed using a CAD software. After all the layers are cut, they must be aligned. A common method for the alignment is the use of alignment holes that are at the same spot in every layer. In the final step, the aligned layers must be bound together. The chosen bonding method is responsible for the pressure resistance of the microfluidic device. A bonding method that works with nearly all materials is the use of adhesive [92]. Unfortunately, it is prone to break upon high pressure (>5 bar), uneven bonding and the risk of channel clogging by residues of the adhesive. In contrast, thermal bonded layers can withstand higher pressures [92]. In this method, the layers are melted together at temperatures near the glass transition temperature. A disadvantage of this method is that unwanted deformation and bubbles can occur when the layers cool down. An alternative method for the manufacturing of microfluidic devices is molding. Molding-based methods can be divided into replica molding, injection molding and hot embossing [93]. Especially for the fast design of prototypes, replica molding is the method of choice and will, therefore, be discussed in detail in this chapter. For replica molding, a template (also called master or negative form) must be manufactured that contains the negative structures of the microfluidic chip (see Figure 3) [94]. The negative structures can be created by photolithography or high precision micro-milling (HPMM). Two common materials for the photolithography are silicones or photoresist materials (e.g., SU-8). For HPPM, harder materials, such as brass, are suitable. A more modern method for creating the template is nanoimprint lithography (NIL) [95]. This method achieves accuracies in the nanometer range. After the template is manufactured, the material for the microfluidic device is poured on it and cured. The standard material for this purpose is PDMS [96]. Due to its low surface energy, PDMS can enter even into structures down to 0.1 µm and is easily removable from the template once it is cured. Additional features of PDMS are its hydrophobicity, good biocompatibility, optical transparency, high elasticity, and low price. Therefore, it is used as the standard material for replica molding of microfluidic devices [97]. After separating the molded target structure from the template, open channels are left in the material. To seal these structures, multiple molded layers can either be stacked on top of each other to generate three-dimensional structures or the open structures can be closed by bonding the layer to a glass layer (e.g., anodic bonding). The latter method allows the direct visual tracking of the fluids. Multiple methods are available to bind stacked layers. With two PDMS layers, it is possible to add a saturated base on one layer and saturated curing agent on the other to form an irreversible connection. Additionally, the binding can be achieved via thermal or plasma treatment. The combination of photolithography and molding with PDMS results in accuracies of appx. ± 10 µm. Due to the elastomeric character of PDMS, this method is often referred to as soft lithography in the literature. Using 3D printing to manufacture microfluidic devices is another option. Three-dimensional printing enables the fast, and cost-effective creation of three-dimensional structures in just one step. In general, 3D printing processes can be divided into extrusion-based manufacturing, stereolithography (SLA) and inkjet printing. Furthermore, there are four approaches to the printing process. In the direct approach, the microfluidic device is printed completely alone by a 3D printer. In the mold-based approach, the template for the molding process is created with the help of a 3D printer. This can replace the conventional photolithographic process. By using a 3D printer, no additional expensive equipment is needed, and manufacturing time can be reduced. In the hybrid approach, only some parts, such as the channels, are created using 3D printing. These parts are then bound to classic materials (e.g., PDMS or glass), combining a quick manufacturing process with the beneficial properties of the standard materials, such as optical transparency. The continuous improvement in 3D printing methods has led to high accuracy manufacturing of units, such as the 15 µm × 15 µm valve, printed by Sanchez Noriega et al. [98]. As 3D printing of microfluidic devices is a highly dynamic field in which several working groups are active, the reader is referred to current review papers and books for a more detailed overview, e.g., [99,100,101]. Microfluidic devices can additionally be made from chromatographic paper. The so-called paper analytical devices (PADs) do not require any pumps due to their ability to transport the fluid via capillary forces. With a modified inkjet printer, wax can be applied in customizable patterns. The wax can penetrate the paper fibers and builds a hydrophobic barrier [82]. Lee et al. describe that different ligands can be coupled to selectively bind target structures [102]. Examples for PADs are pregnancy tests and more recently, coronavirus tests [103,104].

### 3.2. Structures and Modifications for Microfluidic Devices

For the design of microfluidic devices that are usable for the separation of exosomes, several structures need to be implemented. These structures can be divided into general structures, which are present in all microfluidic devices and specialized structures for the realization of a certain separation principle. The main general structure in a microfluidic device is a channel. In their diameters, channels of microfluidic devices can range from the nanometer scale up to hundreds of micrometers. The accuracy of the channel diameter and its surface structure is strongly dependent on the manufacturing method. If possible, round profiles should be preferred over rectangular profiles due to their lower hydraulic resistance [105]. The arrangement and modification of the microchannels allows various fluid manipulations. Passive hydrodynamic mixing can be realized by implementing specially designed channel routes (e.g., 3D serpentines) or by adding obstacles, such as pillars, into the flow path [106]. Valves are essential units to control fluid pathways [107]. They can be realized with two intersecting channels in different layers. Therefore, one channel is placed in a pneumatic layer and the other one in the main layer above or below. When the pressure in the pneumatic channel increases, the elastic material is pressed into the other channel and blocks it [82]. Multiple other microfluidic units for general operations, such as detectors, flow chambers and pumps, have been developed in recent years [108].

#### 3.2.1. Design of Physical Property-Based Microfluidic Devices

It is not only affinity separation that can benefit from implementation into a microfluidic device, but other separation methods can be adapted into the microfluidic world. One example is a microfluidic device for the separation based on the physical properties of exosomes [45]. These methods are sometimes referred to as label-free separation methods compared to affinity-based methods, which are usually annotated as label-based. Strictly speaking, this terminology is not correct, as it does not consider affinity methods in which no ligands are bound to the final purified exosomes. Their main advantage is the reduced risk of the contamination of the purified exosomes with ligands. One of the physical property-based methods is microfluidic filtering [109]. Liu et al. utilized this technique by developing a microfluidic device (ExoTIC) that contained a nanoporous membrane [110]. The simple filtration enabled a separation of exosomes from plasma, urine, and lung bronchoalveolar lavage fluid. Another method based on the physical properties is called deterministic lateral displacement (DLD), describing the effect when fluids containing nanoparticles are pushed through a narrow pillar array. Hattori et al. designed a device with micropillar (diameter: 2.4 µm and gap: 2.6 µm) and nanopillar (diameter 0.47 µm and gap: 1.0 µm) arrays to successfully separate exosomes [111]. Liu et al. designed a microfluidic device with straight microchannels with a high-aspect ratio (height: 50 µm and width: 20 µm) for viscoelastic flow sorting of exosomes. This separation method is based on the “particle migration caused by size-dependent elastic lift forces in a viscoelastic medium” [112]. To achieve a highly viscoelastic fluid character, they added 0.1% polyoxyethylene (POE) to a serum and cell culture media sample. To apply external force field-based methods into microfluidics, specialized units have to be implemented in the microfluidic device. One example is the alternating current electrokinetic (ACE) microarray chips developed by Ibsen et al. [29]. It utilizes the dielectrophoretic (DEP) force generated with a platinum electrode to accumulate exosomes in DEP high-field regions. Another example is the integration of two sequential surface acoustic wave (SAW) microfluidic modules by Wu et al. [113]. These modules were able to create an acoustic radiation force, which deflected blood cells in the first module and MVs and apoptotic bodies in the second module towards a waste channel, and therefore enabled the separation of exosomes. A more detailed overview of physical-based separation methods was recently published by Hassanpour Tamrin et al. [52].

#### 3.2.2. Design of Affinity-Based Microfluidic Devices

The high selectivity of affinity binding has led to the development of special designs in microfluidic devices to maximize the benefits of this separation principle. One approach for affinity-based separation of exosomes is the immobilization of the ligands on the inner surfaces of these devices. This review focuses on polymers as manufacturing material, because of their advantageous features for the manufacturing process. In general, the polymer surface should be altered to enhance wetting, and thereby reduce air bubble formation [114]. For affinity separation, the modification of the inert surface is essential to enable the binding of ligands. In the best case, the coupled ligands should be able to withstand the wall shear stress at higher flowrates and should not detach during storage. Therefore, covalent binding of the ligands to the surface is preferred [115]. To achieve this, it is important to know the chemical composition of the material. PDMS consists of silicon atoms to which two methyl groups are coupled. The cross-linked structure of PDMS is created via siloxane bonds. This results in the overall inert and hydrophobic properties of the material [116]. To enable the covalent coupling of ligands, hydrocarbon groups can be removed by means of oxygen plasma treatment. Afterwards, silanol or epoxy groups can bind to the oxidized PDMS surface. Following the silanization step, ligands (e.g., aptamers) can covalently bind to the surface (Figure 4). Finally, a blocking step must be carried out (e.g., with ethanolamine) to prevent non-specific binding. Instead of oxygen, Shakeri et al. presented a method using carbon dioxide plasma [117]. This treatment generates hydroxyl and carboxyl groups on the surface. These chemical groups can directly bind to amino acids without further treatment. A chemical method for binding antibodies to PDMS surfaces was used by Hisey et al. [118]. They first incubated the PDMS channels with 10% [3-(2-aminoethylamino)-propyl]-trimethoxysilane in ethanol. This was followed by another incubation step with 5% glutaraldehyde in distilled water. After a washing step with distilled water, the treatment made it possible to covalently bind anti-EpCAM and anti-CD9 antibodies to the channel surfaces. Nair et al. described in detail the modification of hot embossed COC to couple a ssDNA linker that contained uracil [119]. Again, the surface must be activated first. Nair et al. used UV radiation (254/185 nm) for 15 min, with a power input of 22 mW cm^−2^ in combination with ozone. This leads to the formation of carboxyl groups on the surface, which are able to covalently bind to the 5′-terminus of the ssDNA linker. This linker can then be modified with specific antibodies. The coupling technique was recently used by Wijerathne et al. to manufacture a microfluidic device for the enrichment of EVs to diagnose acute ischemic strokes [120]. The uracil group contained in the linker enables simple elution after capture, as it can be digested by the addition of enzymes. An additional overview table with chip substrates and their fabrication methods for affinity separations can be found in the review paper by Bao et al. [108].

#### 3.2.3. Enhancing the Affinity Separation Effect in Microfluidics

To enhance the affinity separation effect, surface area and flow conditions are critical parameters [121,122]. Therefore, multiple specialized flow routes and structures have been developed. An example of such structures are long beds filled with micropillar arrays, which decrease the diffusion distance and provide higher residence time for the affinity binding process [109]. This method was used by Wijerathne et al., who tested two types of micropillar arrays [120]. One array contained circular pillars, with a diameter of 100 µm and 15 µm spacing made from COC and another with diamond shaped pillars, with a 10 µm × 10 µm square base and 10 µm spacing made from COP. Both were functionalized with anti-CD8α antibodies coupled to oligonucleotide bi-functional linkers. The different number of parallel beds, and therefore number of pillars, made a comparison difficult. However, the diamond-shaped columns showed significantly higher flow rates, with no change in recovery. Another structure was evaluated by Hisey et al. [118]. They used a herringbone grooved surface made from PDMS that was functionalized with anti-CD9 or anti-EpCAM antibodies. The herringbone structures had a V-like shape with a length of 400 µm for one branch of the “V” and 135 µm the other, connected in a 45° angle (see Figure 5). The positive herringbone grooves enhanced the mixing of the sample, and therefore improved the contact of the solution with the antibodies. A further way to increase the surface area and enhance the mixing was presented by Zhang et al. [123]. They manufactured a 3D porous serpentine nanostructure via patterned colloidal self-assembly. Briefly, 1 µm silica beads were placed into microfluidic chambers and then connected with (3-mercaptopropyl)-trimethoxysilane (3-MPS). This created a closely packed microbead pattern with appx. 150 nm pores. The microbeads were then activated with 4-maleimidobutyric acid N-hydroxysuccinimide ester (GMBS) and afterwards, functionalized with anti-CD81 antibodies. This method allowed the capture of exosomes from diluted plasma samples. Kang et al. utilized multiple circular chambers (see Figure 2) in their approach to increase the contact surface [34]. Their new ExoChip consisted of 30 parallel rows, with 60 connected chambers with a diameter of 500 µm [34]. Unlike their previous device (ExoChip), which had anti-CD63 antibodies coupled as ligands, the new device has immobilized annexin V. Annexin V can bind to phosphatidylserine (PS), which, as recent evidence suggests, is overexpressed on the surface of cancer-derived exosomes [34]. The new device showed significantly higher capture efficiency in comparison to the older device, highlighting the potential of the discovery of new targets on exosomes.

#### 3.2.4. Microfluidic Design for Affinity Approaches Utilizing Beads

Beads coupled with affinity ligands have already shown good separation results in conventional separation techniques for exosomes [124,125,126]. Microfluidic separation can enhance these benefits for both non-magnetic and magnetic beads. The use of beads can also reduce the effort of the manufacturing process, by eliminating the necessity to immobilize ligands on the surfaces of the devices. One example for this is the device from Tayebi et al. [127]. They used non-magnetic microparticles with biotin-streptavidin-bound anti-CD63 antibodies for hydrodynamic trapping of exosomes. After binding the exosomes, the beads were trapped in hydrodynamic pockets during their way through the device, thus reducing the complexity of the manufacturing process. Another approach was developed by Xu et al. [128]. They combined the design of microfluidic flow routes for enhanced mixing with magnetic affinity nanoparticles. Streptavidin-coated magnetic beads were used, which were functionalized with biotinylated-mouse-Tim4-Fc. This structure can bind to PS on exosomes. To enhance the binding between exosomes and magnetic beads, the microfluidic device was manufactured with Y-shaped micropillar arrays (see Figure 5). Comparative experiments led to optimal micropillar spacing of 50 µm and a flow rate of 0.2 µL min^−1^. By placing or removing a permanent magnet under the micropillar area, the magnetic beads were retained or eluted. Due to the Ca^2+^-dependency of the bond between Tim4-Fc and PS, the affinity binding can easily be dissolved by adding a chelating agent. This enables non-disruptive, quick enrichment of label-free exosomes. These examples show that smart combinations of different separation principles can enhance the success of microfluidic separations, while reducing the complexity of the manufacturing process.

## 4. Bioassays—Validation of the Separation

After designing a microfluidic device for the separation of exosomes, it is necessary to analyze and validate the results of the separation. Determination of the size distribution and quantification of the separated exosomes can be carried out by utilizing nanoparticle tracking analysis (NTA). A disadvantage of NTA is its unspecificity towards exosomes. Thus, the quantification process can be interfered with by lipoproteins and protein aggregates. To overcome the unspecificity, multiple enzyme-linked immunosorbent assays (ELISA), fluorescent assays [26], surface-enhanced Raman scattering (SERS) approaches [129] and nanoflow cytometry setups [130] have been developed. In addition, novel methods, such as the use of electrokinetic detection with a microcapillary [131] and droplet digital ELISAs (ddExoELISA) [132], show great potential to simplify the detection process and enhance the detection limit. As already mentioned, exosomes carry proteins and RNA fragments, which can be used as prognostic and diagnostic markers for several diseases. To analyze the cargo, captured exosomes must be lysed first. Western blot (WB) analysis, sodium dodecyl sulfate polyacrylamide gel electrophoresis (SDS-PAGE) and nano liquid chromatography mass spectrometry (nano LC–MS) can deliver insights into the proteome and transcriptome of the captured exosomes [7]. Another specialized method to analyze the RNA cargo is (real time) quantitative polymerized chain reaction (qPCR). When developing a new device, it can be feasible to optically inspect the microfluidic device and the exosomes to evaluate the performance of the separation. This can be performed by scanning electron microscopy (SEM) or (bio-)transmission electron microscopy (TEM). The design of sensors and detectors is a large interdisciplinary field. An overview of the techniques to analyze exosomes is shown in Figure 6. For LOC and µTAS devices, the development of transportable detectors that can be implemented into microfluidic devices has accelerated in recent years. Since this topic is beyond the scope of this work, the reader is referred to current reviews, e.g., [74,133]. However, we want to also highlight the recent advances for extracellular vesicle detection and sensing. The newest technologies for efficient sensing include refractive index-based sensors and magneto-electrochemical sensors [134,135]. A great overview of the different sensing technologies, ranging from plasmon resonance, scattering, ELISA and electrochemical-based assays, is given by Im et al. [136]. Current commercially available kits for exosome isolation have been reviewed very recently by Shirejini and Inci [60]. They highlight the advantages and disadvantages of commercially available kits, such as high-throughput, cost-effectiveness, time-effectiveness and others. They review membrane-based methods, precipitation-based methods, size-exclusion-based methods and immune-affinity-based methods [60]. The commercially-available products are as follows: ExoMir (Bioo Scientific) as the membrane-based method; EXO-Prep (Hasna BioMedLife Sciences), Exosome Purification Kit (Norgen Biotek), Exo-Spin Isolation Kit (Cell Guidance Systems), ExoQuick Exosome Precipitation (System Biosciences), PureExo Exosome Isolation Kit (101 Bio), miRCURY Exosome Isolation Kit (Exiqon), Total Exosome Isolation Reagent (Invitrogen), Minute High-Efficiency Exosome Precipitation Reagent (Invent Biotechnologies), RIBO Exosome Isolation Reagent (RIBO) as the precipitation-based methods; qEV (iZON Science), EVSecond (GL Sciences), ExoLutE (Rosetta Exosome Company), PURE-Evs (HansaBioMed) as the size-exclusion-based methods; Exosome-Human EpCAM isolation reagent (Thermofisher), Exosome Isolation Kit CD81/CD63 (Miltenyi Biotec), Exosome Isolation and Analysis kit (Abcam), MagCapture Exosome Isolation Kit PS (FUJIFILM Wako Pure Chemical Corporation) as the immune-affinity-based methods; ExoEasy Maxi Kit (Qiagen) and Capturem Exosome Isolation Kit (Takara Bio) as the affinity-spin-column-based methods.

## 5. Conclusions

To exploit the full potential of exosomes for prognostics, diagnostics and theranostics, their complex separation must be mastered first. The diversity of biofluids in which exosomes can be found, as well as their high heterogeneity as they are derived from different donor cells, imposes a challenge regarding the separation process. Exosome separation methods that have already been developed and are, in some cases, already commercially available do not provide sufficient answers to every separation application yet [1]. For example, physical separation methods can be disturbed by very similar LP impurities. In addition, conventional methods are often reduced to a well-equipped laboratory, and thus prevent the application of exosomes in a POC scenario [85]. This review highlights how the impurity issues might be overcome by highly specific affinity separation. The possibilities of affinity separation come with the price that this method cannot be developed according to a simple scheme, but requires several manual optimization steps to achieve the desired separation, e.g., the selection and design of a ligand and the determination of the dissociation constants. The application limitations of exosomes can be solved by the straightforward manufacturing of microfluidic devices [99]. These devices are cheap in production and can work with low sample volumes in an unclean environment using minimal to no additional equipment. Thus, combining the affinity-based separation methods with microfluidic devices has great potential for POC diagnostics. Using state-of-the-art production methods, such as 3D printing, complex geometries can be realized in microfluidic devices. We also want to highlight 3D-printing technologies as very fast and cost-efficient technologies that can be used to generate ideal options for small EV isolation. Adapting these micrometer-scale structures to a specific target ligand system can additionally be used to optimize the residence time. Further innovations and optimizations in designing improved structures that modify the flow will emerge with future research on fluid dynamics. In addition, other conventional methods can be adapted to the microfluidic world and combined with affinity separation—thus contributing to further progress. Another contribution to the upcoming improvements can be provided by new insights into the function, physiology, and pathophysiology of exosomes inside the human body. As new target structures and molecular binding partners are discovered, new ligands come into focus that have higher selectivity or better dissociation constants than the ones used in the current devices. This progress will inevitably lead to the development of a variety of innovations in this field. However, with the toolbox of methods shown here, this development does not have to start from scratch, but can build on the state-of-the-art modular manufacturing approaches.

## Figures and Tables

**Figure 1 ijms-23-09004-f001:**
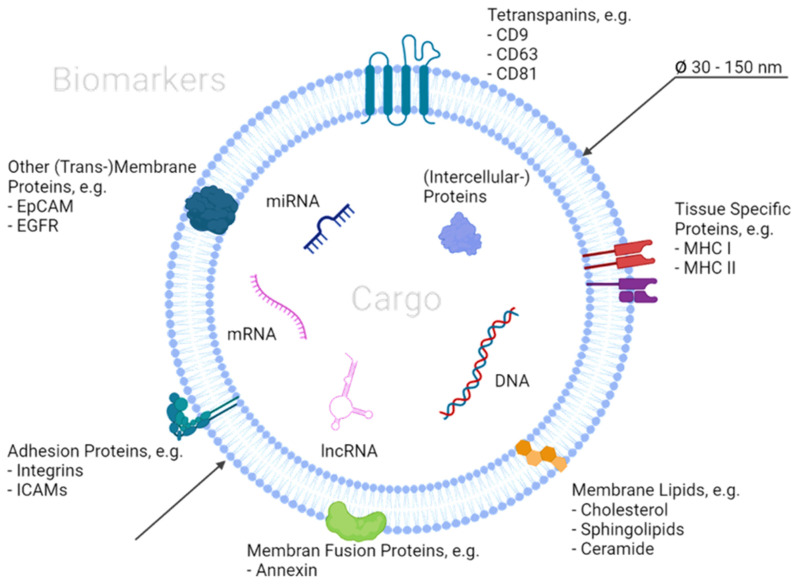
Schematic representation of an exosome with surface and cargo biomarkers. While the surface biomarkers are promising targets for affinity separation methods, the cargo can be used for exosome characterization. Biomarkers can roughly be categorized into adhesion, fusion and recognition proteins, as well as membrane lipids.

**Figure 2 ijms-23-09004-f002:**
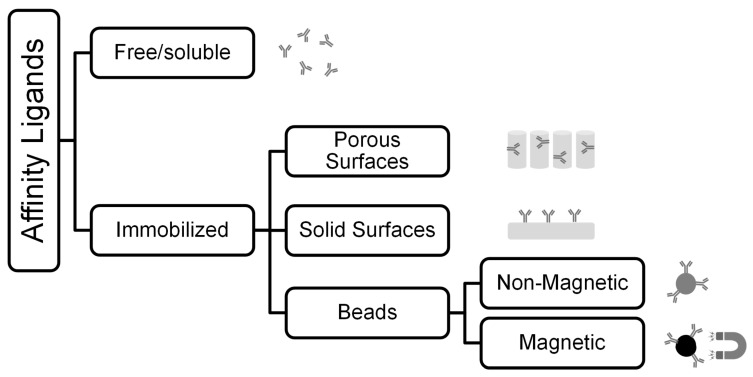
Overview of different possibilities to use ligands for affinity separation.

**Figure 3 ijms-23-09004-f003:**
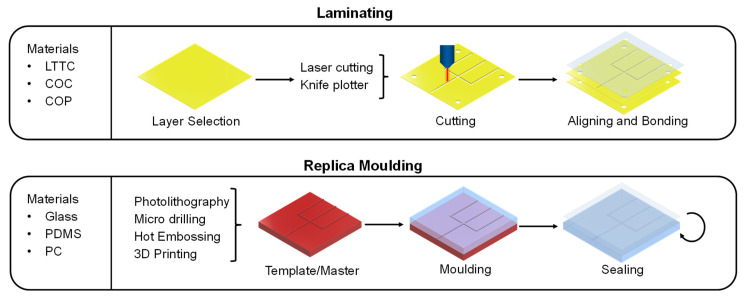
Overview of the two manufacturing processes laminating and replica-molding for microfluidic devices.

**Figure 4 ijms-23-09004-f004:**
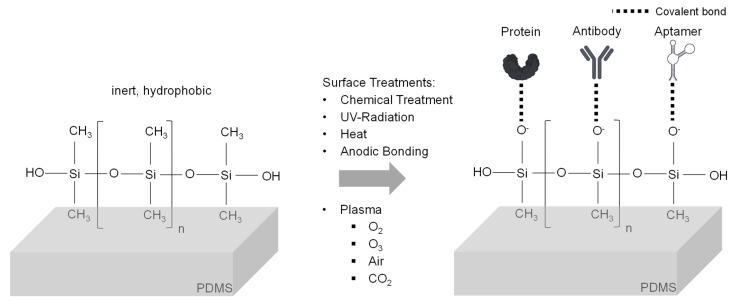
Activation of PDMS surface and ligand coupling.

**Figure 5 ijms-23-09004-f005:**
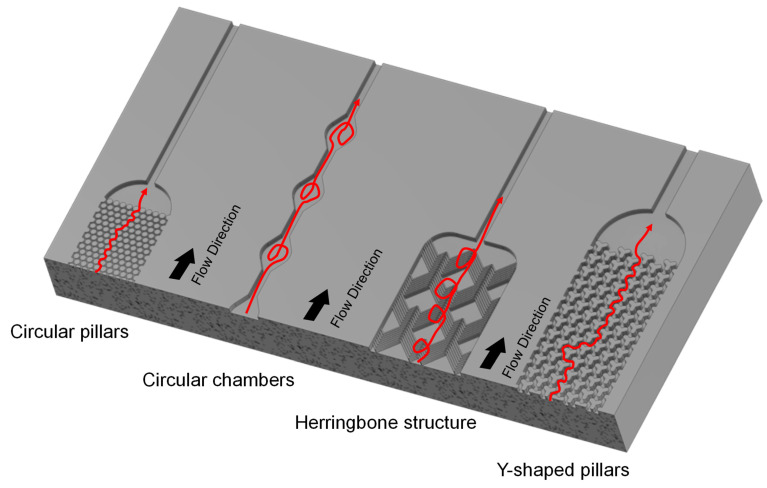
Comparison of four different design concepts to enhance the performance of microfluidic affinity separations. The different structures increase the contact surface, improve the mixing, and therefore optimize the binding reaction of immobilized ligands with exosome surface markers. The structures are derived from the literature (presented in the text) and were re-constructed true to scale in Autodesk Inventor Professional in one part for better comparison. Flow direction of the sample is indicated next to the structures.

**Figure 6 ijms-23-09004-f006:**
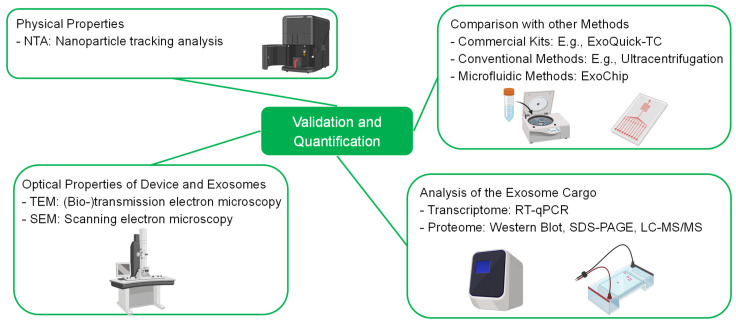
Validation and quantification with different analysis methods.

**Table 1 ijms-23-09004-t001:** Surface biomarkers of exosomes from different biofluids and cancer types [50].

Cancer Cell Type	Biofluid	Biomarkers
Bladder	Whole blood	CD9, CD81
Bladder	Urine	CD63
Bladder and liver	Plasma	CD63
Liver	Serum	CD63
Lung	Plasma	CD9, CD63, CD81, CD82, CA125, EpCAM, IGF-1R
Prostate	Serum	EpCAM
Breast	Serum	PSA
Ovarian and breast	Plasma	CD24, EpCAM, FRα, HER2, MMP14
Ovarian	Plasma	CD9, CD63, CD81, EpCAM
Ovarian	Ascites	CD24, EpCAM
Glioblastoma multiforme	Blood	CD63, EGFR, EGFRvIII, EPphA2, IDH1, PDGFR, PDPN, R132H
Melanoma	Whole blood	CD9
Colorectal and gastric	Ascites	EV glycans

**Table 2 ijms-23-09004-t002:** Surface biomarkers of exosomes from different immune cells [51].

Immune Cell Type	Biomarkers
Dendritic cells	MHC complexes, CD89, CD86, ICAM-1
B cells	D19, CD37, MHC II
T cells: Treg	D25, CD73, CTKA-4
T cells: CD4+	CD4, HLA class 1, TfR, TCR, integrin β2, Fas ligand
T cells: CD8+	CR Fas ligand CD8
Natural killer cells	D56, perforin, granzyme B, Fas ligand
Macrophages	Bacterial antigens, cofilin-1, GNB1, actin, cyclophilin A
Mast cells	MHC II, c-Kit, LFA1, ICAM

## Data Availability

Not applicable.

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
