# Peer review of "Microfluidic Approaches for Affinity-Based Exosome Separation"

_ijms, 2022, doi:10.3390/ijms23169004_

Round 1

Reviewer 1 Report

The Authors review the microfluidic Approaches for Affinity-Based Exosome Separation. The manuscript is well written and it could be useful for the readers to approach the topic.  Here, my comments to the manuscript:

- the quality of the images should be improved;

- an overview of the innovative exosome sensing devices could be reported in order to rate the exosome separation (see, e.g. Highly Sensitive Refractive Index Sensor Based on Polymer Bragg Grating: A Case Study on Extracellular Vesicles Detection. Biosensors12(6), 415, 2022; Novel nanosensing technologies for exosome detection and profiling. Lab on a Chip17(17), 2892-2898, 2017; J Integrated magneto–electrochemical sensor for exosome analysis. ACS nano10(2), 1802-1809, 2016).

Author Response

The Authors review the microfluidic Approaches for Affinity-Based Exosome Separation. The manuscript is well written and it could be useful for the readers to approach the topic.  Here, my comments to the manuscript:

- the quality of the images should be improved;

The quality of the images has been improved (we changed text and colourings to improve the readability of the pictures) (However, it was not clear what kind of improvement you wanted to indicate and which picture you mean specifically)

- an overview of the innovative exosome sensing devices could be reported in order to rate the exosome separation (see, e.g. Highly Sensitive Refractive Index Sensor Based on Polymer Bragg Grating: A Case Study on Extracellular Vesicles Detection. Biosensors12(6), 415, 2022; Novel nanosensing technologies for exosome detection and profiling. Lab on a Chip17(17), 2892-2898, 2017; J Integrated magneto–electrochemical sensor for exosome analysis. ACS nano10(2), 1802-1809, 2016).

All of those studies you are suggesting here are definitely of great value and impact. We included the mentioned publications and discussed them in the bioassays section of the manuscript.

Thank you for your time and your consideration helping us to improve the manuscript.

Reviewer 2 Report

In this article, Theel and colleagues investigate the microfluidic approaches for affinity-based exosome separation. The article looks nice, demonstrating that the authors spent much time preparing the manuscript. There exist some comments that require careful attention.  

1.       The term exosome is recommended not to be used anymore by the ISEV. The committee recommended using small EVs (sEVs) instead of that, so the reviewer recommends applying such a comment within the manuscript.

2.       In Table 1, markers like CD63 or CD9 were known to be the common surface marker of exosomes. How Bladder in Urine Biofluid doesn’t express other common surface markers?

3.       A high quality of Fig. 1 must be included and replaced in the manuscript.

4.       Authors are recommended to provide a paragraph and discuss commercially available kits and elaborate on their pros and cons.

5.       In the conclusion of the article, it would be great to see if the authors can recommend an “ideal” microfluidic platform for the isolation of small EVs. Indeed, what modifications to currently available designs are required to be an ideal option for small EV isolation? 

Author Response

In this article, Theel and colleagues investigate the microfluidic approaches for affinity-based exosome separation. The article looks nice, demonstrating that the authors spent much time preparing the manuscript. There exist some comments that require careful attention.  

  1. The term exosome is recommended not to be used anymore by the ISEV. The committee recommended using small EVs (sEVs) instead of that, so the reviewer recommends applying such a comment within the manuscript.

We added this comment in the introduction. Thank you for pointing this out.

  1. In Table 1, markers like CD63 or CD9 were known to be the common surface marker of exosomes. How Bladder in Urine Biofluid doesn’t express other common surface markers?

There should also be other surface markers such as Uroplakin 2 and 3A which can be found in sEVs and used for detection. However, CD63 is the most common one. If you think more markers need to be included, I am very happy to include them in this table.

  1. A high quality of Fig. 1 must be included and replaced in the manuscript.

The quality of Figure 1 has been improved.

  1. Authors are recommended to provide a paragraph and discuss commercially available kits and elaborate on their pros and cons.

 This is a very good point. We added this to the Bioassay section.

  1. In the conclusion of the article, it would be great to see if the authors can recommend an “ideal” microfluidic platform for the isolation of small EVs. Indeed, what modifications to currently available designs are required to be an ideal option for small EV isolation? 

Well, I think that an ideal system depends on multiple parameters. I am a great fan of 3D-printing technology and think that here we have the most cost-efficient solutions for small EVs isolation. Thus, we added a sentence in our conclusion.

Thank you very much for your time and consideration of our manuscript.

Reviewer 3 Report

The manuscript by Theel and Schwaminger provides a good review of affinity-based microfluidic separation of exosomes. The authors have provided a good discussion on the theory behind affinity separation, which is very useful for the reader. The manuscript is well written and very informative. Therefore, I recommend publishing this review article without significant changes. However, the authors must correct the error in line 312, where Keq,d has to be 10-15 M. Authors should also mention the anodic bonding that is utilized in glass devices.

Author Response

The manuscript by Theel and Schwaminger provides a good review of affinity-based microfluidic separation of exosomes. The authors have provided a good discussion on the theory behind affinity separation, which is very useful for the reader. The manuscript is well written and very informative. Therefore, I recommend publishing this review article without significant changes. However, the authors must correct the error in line 312, where Keq,d has to be 10-15 M. Authors should also mention the anodic bonding that is utilized in glass devices.

Thank you very much for your time and your comments. We corrected the error and wrote the exponent in superscript.

We also added anodic bonding as being used for glass layers.

Round 2

Reviewer 2 Report

Authors addressed all my comments.